# Modulation of Dyslipidemia Markers Apo B/Apo A and Triglycerides/HDL-Cholesterol Ratios by Low-Carbohydrate High-Fat Diet in a Rat Model of Metabolic Syndrome

**DOI:** 10.3390/nu14091903

**Published:** 2022-05-01

**Authors:** Abrar Alnami, Abdulhadi Bima, Aliaa Alamoudi, Basmah Eldakhakhny, Hussein Sakr, Ayman Elsamanoudy

**Affiliations:** 1Clinical Biochemistry Department, Faculty of Medicine, King Abdulaziz University, Jeddah 21465, Saudi Arabia; AbrarAlnami@outlook.com (A.A.); hadibima@hotmail.com (A.B.); aliaa.alamo@gmail.com (A.A.); beldakhakhny@kau.edu.sa (B.E.); 2Physiology Department, College of Medicine and Health Sciences, Sultan Qaboos University, Muscat 123, Oman; sakr_doctor@yahoo.com; 3Medical Physiology Department, Faculty of Medicine, Mansoura University, Mansoura 35516, Egypt; 4Medical Biochemistry and Molecular Biology Department, Faculty of Medicine, Mansoura University, Mansoura 35516, Egypt

**Keywords:** metabolic syndrome, dyslipidemia, HFLC, ketogenic diet, HOMA-IR, Apo B/Apo A ratio, TG/HDL ratio

## Abstract

Metabolic syndrome (MetS) risks cardiovascular diseases due to its associated Dyslipidemia. It is proposed that a low-carbohydrate, high-fat (LCHF) diet positively ameliorates the MetS and reverses insulin resistance. Therefore, we aimed to investigate the protecting effect of the LCHF diet on MetS-associated Dyslipidemia in an experimental animal model. Forty male Sprague-Dawley rats were divided into four groups (10/group): the control group, dexamethasone-induced MetS (DEX) (250 µg/kg/day), LCHF-fed MetS group (DEX + LCHF), and High-Carbohydrate-Low-Fat-fed MetS group (DEX + HCLF). At the end of the four-week experiment, fasting glucose, insulin, lipid profile (LDL-C, HDL-C, Triglyceride), oxidized-LDL, and small dense-LDL using the ELISA technique were estimated. HOMA-IR, Apo B/Apo A1 ratio, and TG/HDL were calculated. Moreover, histological examination of the liver by H & E and Sudan III stain was carried out. In the DEX group, rats showed a significant (*p* < 0.05) increase in the HOMA-IR, atherogenic parameters, such as s-LDL, OX-LDL, Apo B/Apo A1 ratio, and TG/HDL. The LCHF diet significantly improved the parameters of Dyslipidemia (*p* < 0.05) by decreasing the Apo B/Apo A1 and TG/HDL-C ratios. Decreased steatosis in LCHF-fed rats compared to HCLF was also revealed. In conclusion, the LCHF diet ameliorates MetS-associated Dyslipidemia, as noted from biochemical results and histological examination.

## 1. Introduction

Diet is a keystone of any lifestyle intervention program. The most common dietary plan strategy restricts energy, and based on that, several other nutritional methods have been proposed, such as Low-Carbohydrate High-Fat diets (LCHFD) [1]. According to the Dutch Food Consumption Survey (FCS, 2007–2010), the daily requirements of macronutrients are recommended to be as follows: 45% carbohydrates (21% simple sugars and 24% complex sugar, including starch), 35% fats, 15% proteins, and 5% fibers. On that basis, a study assumes that the term “low” could refer to lower than 45% derived from carbohydrates [2], whereas another study assumed LCHFDs allow 20 to 60 g/d and Very-Low-Carbohydrate Ketogenic Diets (VLCKDs) typically restrict carbohydrates to less than 20 g/d [3]. There has been growing interest in Low-Carbohydrate/High-Fat diets in recent years because it is proposed to have been associated with positive impacts, such as weight reduction, enhanced insulin sensitivity, reasonable glycemic control in pre-diabetics and diabetics, reduction in cardiovascular disease risk factors, and increased feelings of satiety [2,4].

Metabolic syndrome (MetS) is a cluster of a group of disorders that characterize significant risk factors for cardiovascular disease (CVD) and type 2 diabetes mellitus [5]. These factors are visceral obesity, elevated blood pressure, elevated triglyceride level, reduced HDL-Cholesterol, and high blood sugar [6]. Atherogenic Dyslipidemia is a crucial component of metabolic syndrome related to coronary atherosclerosis. The starring role of plasma triglycerides (TG), low-density lipoprotein cholesterol (LDL-C), and high-density lipoprotein cholesterol (HDL-C) in the pathogenesis of coronary heart disease (CHD) have been well documented [7]. Today’s central debate is whether LDL-C should continue as the primary adjustable for assessing CV risk and targeting lipid-lowering therapy [8]. At present, it is suggested that other lipid parameters may better represent the risk of CHD other than elevated LDL-C and conventional markers, such as oxidized-LDL (ox-LDL) and small dense-LDL (sd-LDL) [9].

Dyslipidemia is a metabolic disorder involving abnormal blood lipid levels [10]. It is reported that Dyslipidemia is a genetically predisposed disorder, and much of the population acquire it from multifactorial reasons, such as obesity, diet, and lifestyle habits [11]. Dyslipidemia is closely related to insulin resistance, resulting in increased triglyceride-rich lipoproteins (TRLs), decreased HDL-C, and increased sd-LDL/ox-LDL particles [12].

It has been proposed that the Low-Carbohydrate/High Fat (LCHF) diet has a positive effect on metabolic-syndrome-related dyslipidemia. Therefore, the present study aims to study the possible protecting role of the Low-Carbohydrate/High-Fat diet on Dyslipidemia in an experimental animal model of metabolic syndrome.

## 2. Material and Methods

The current study is an experimental animal study. It was conducted in the Department of Clinical Biochemistry-Faculty of Medicine—King Abdulaziz University, Jeddah, Saudi Arabia. The Biomedical Ethics Research committee, Faculty of Medicine, King Abdulaziz University, Jeddah, Saudi Arabia approved the study (Reference No 33-21).

### 2.1. Animals and Experimental Protocol

Forty male Sprague-Dawley rats (120 ± 20) grams were obtained from the Animal House at King Fahd Medical Research Center, Jeddah, Saudi Arabia, and enrolled in this study. The animal study was carried out following the animal welfare act and guide for care use of Experimental Research Center, King Fahd Medical Research Center, King Abdulaziz University. The experiment was approved by The Animal Care and Use Committee (ACUC) at King Fahd Medical Research Center (Protocol Number: ACUC-20-09-21), King Abdul-Aziz University, Jeddah. Animals were housed in metal cages with meshes (temperature: 24 °C ± 3 °C; humidity: 40–70%, and 12/12 h light/dark cycle). They fed on ad-libitum standard laboratory rat chow, and they had free access to drinking water for the first ten days to allow acclimation to the new environment.

The rats were grouped into four groups (10 rats/each). According to the research protocol, each group has its type of diet: Group I (Control) served as a negative control group, included ten Albino Wistar rats with regular diet, and saline was injected subcutaneously. Rats in the other three groups were injected with dexamethasone subcutaneously at a dose of 250 µg/kg/day dissolved in saline [13]. Group II (DEX): continued on the chow diet. Group III (DEX + LCHF): rats were fed with a Low-Carbohydrate, High-Fat diet for four weeks. Group IV (DEX + HCLF): rats were fed on a High-Carbohydrate Low-Fat Diet for four weeks.

### 2.2. Drug and Dosage

Metabolic syndrome was induced by dexamethasone injection, which was obtained from Saudi Pharmaceutical Industries (SPI), Riyadh, Saudi Arabia. It was injected into rats subcutaneously at 250 µg/kg single dose daily, according to the method described by Sivabalan et al. [13].

### 2.3. Rats Feeding

The three types of diet used are isocaloric. The LCHF diet comprises 85 kcl% fat, 5 kcl% carbohydrates, and 10 kcl% protein. The HCLF contains 85 kcl% carbohydrates, 10 kcl% fat, 5 kcl% protein. LCHF and HCLF diets were customized from Research Diets D10070801, New Brunswick, NJ, USA. The diet composition and preparation were carried out according to Ble-castillo et al. [14]. The detailed description of Chow, LCHF, and HCLF diet compositions is presented in Table 1.

### 2.4. Anthropometric Measurement

Weekly measurements of height, abdominal and thoracic circumference (cm), and body weight (grams) were carried out, and body mass index (BMI) was calculated according to Novelli et al. [15].

### 2.5. Sampling and Biochemical Investigations

Each rat was anesthetized by ether after 12 h of overnight fasting at the end of the experimental protocol duration. The blood samples were collected into plain plastic containers under a complete aseptic condition from retro-orbital venous plexus. Sera were collected after centrifugation at 15,000 rpm for 20 min, then divided into aliquots and stored at −80 °C until the biochemical investigations.

The serum level of fasting glucose (spectrophotometrically, Intertek, London, UK, CS605) and insulin (Rat Insulin ELISA kit, Novus Biologicals, LLC 10730 E. Briarwood Avenue, Building IV, Centennial, CO, USA) for the quantitative measurement of rat insulin in serum and plasma were estimated.

The degree of insulin resistance in each sample was calculated by the homeostasis model assessment of insulin resistance (HOMA-IR) described by Matthews et al. [16]. According to the equation, HOMA-IR was calculated by considering fasting insulin and fasting blood glucose levels: fasting insulin (μU/mL) × fasting plasma glucose (mg/dl) × 0.0551/22.5.

Quantitative determination of rat-serum-oxidized Low-Density Lipoprotein (ox-LDL) (SL0554Ra), Small dense Low-Density Lipoprotein (sd-LDL) (SL-1558Ra), Apo-protein A (SL0817Ra), Apo-Protein B (SL0819Ra), ELISA was performed by Sandwich-ELISA. Serum triglycerides (TG) and HDL-C were measured calorimetrically using kits provided by Intertek, London UK, CS611, and Intertek, London, UK, CS606, respectively. The kits were purchased from SunLong Biotec, Hangzhou, Zhejiang, China. LDL-C was measured by Sandwich-ELISA. LDL ELISA kit was obtained from SunLong Biotec, Hangzhou, Zhejiang, China, SL1637Ra. Triglycerides/HDL-C ratio and Apo B/apo A1 ratio were calculated.

### 2.6. Histological Examination

For the histological study, intracardiac perfusion was performed using 150 mL of buffered paraformaldehyde (4% of 4% buffered paraformaldehyde (pH 7.3). Then, the livers were excised and weighed. Fresh liver tissue from the right lobe was processed for cryosectioning for intrahepatocyte lipid evaluation by Sudan III staining. Frozen sections with 10-μm thickness were stained with Sudan III stain.

The histological grading of hepatic steatosis was graded according to Meli et al. [17], and Abo El-Khair et al. [18]. They used the percentage of hepatocytes that are infiltrated by lipids to determine the degree of steatosis as follows: no infiltration was graded as (0), fatty infiltration of less than 30% of hepatocytes (1); fatty infiltration of 30–70% of hepatocytes (2), and fatty infiltration of more than 70% of hepatocytes (3).

The left lobe was fixed in 10% buffered formalin. Then, the left lobe’s specimens were prepared (dehydrated by alcohol, cleared by xylene, and embedded in molten paraffin) and stained with hematoxylin and eosin (H and E) [19].

### 2.7. Statistical Analysis

Data were analyzed using Statistical Package for Social Science software computer program version 22 (SPSS, Inc., Chicago, IL, USA). Quantitative parametric data were presented as mean and standard deviation. *p*-value ˂ 0.05 was considered statistically significant. The following tests were used: Paired *t*-test for analysis of the anthropometric parameters, One-way Analysis of Variance (ANOVA) and Tukey to compare quantitative parametric data, and Spearman’s correlation was used to correlate the different parameters.

## 3. Results

### 3.1. Anthropometric Results:

The anthropometric parameters of all studied animal groups are presented in Table 2. No difference in height was detected between the animals of each group at all levels of measurement, basal line, during induction of the metabolic syndrome model, and at the post-induction stage. Regarding weight and body mass index (BMI), no difference was detected between the studied group at the baseline stage. At the same time, there was a significant increase in both parameters during the induction and post-induction phase in the DEX group compared to the control group.

In response to the LCHF diet, there was a significant (*p* < 0.05) decrease in weight and BMI versus the DEX group. As expected from the literature [20], the HCLF-diet-fed rats showed a significant (*p* < 0.05) increase in weight and BMI.

### 3.2. Histological Light Microscopic Results

Hematoxylin and eosin-stained liver sections of rats in the control group showed the normal structure of hepatic lobules. The hepatocytes showed regular arrangement with branching and anastomosing cords radiating from a central vein with the portal-tract-contained branches of the portal vein, hepatic artery, bile ducts, and Kupffer cells seen (Figure 1). Tiny lipid droplets were seen (Figure 2). The histological steatosis grade of the control group is represented by 100% grade 0 (Figure 3). Rats from the DEX group showed noticeable histological changes (H&E staining) in the form of disrupted arrangement of the hepatic cords, congested portal veins with cellular infiltration, and irregularly dilated blood sinusoids (Figure 1). In Sudan-III-stained sections, lipid droplets accumulation was evident (Figure 2). The findings of Sudan III staining prove the lipid accumulation. The histological steatosis grades of the DEX-induced MetS group were 0% grade 0, 20% grade 1, 30% grade 2, and 50% grade 3 (Figure 3). The histological findings of the liver sections confirm the development of the metabolic syndrome. It is reported that the occurrence of fatty liver changes is the hepatic manifestation of the metabolic syndrome model. In Sudan-III-stained sections, slight deposition of lipid droplets was detected (Figure 2). The DEX + LCHF diet group showed branching and anastomosing cords of normal hepatocytes radiating from a non-congested central vein and normal non-congested blood sinusoids, which represented the nearly normal structure of hepatic lobules (Figure 1). The findings of Sudan III staining prove the ameliorative effect of the LCHF diet on lipid accumulation. The histological steatosis grades of group K are as follows: 80% grade 0, 10% grade 1, 10% grade 2, and no grade 3 is detected (Figure 3). On the other hand, the DEX + HCLF group showed multiple lipid droplets, dilated congested central vein, dilated congested hepatic sinusoids, and some hepatocytes with dark pyknotic nuclei (Figure 1). In Sudan-III-stained sections, marked deposition of lipid droplets in hepatocytes was distinct (Figure 2), with the histological steatosis grades as: 0% grades 0 and 1, 40% grade 2, and 60% grade 3 (Figure 3).

### 3.3. Biochemical Results

#### 3.3.1. DEX Induces Metabolic Syndrome and Dyslipidemia 

Insulin resistance and Dyslipidemia are evident in the metabolic syndrome group (DEX). Insulin resistance is indicated by increased fasting glucose, plasma insulin levels, and HOMA-IR in the DEX group compared to the control group (Figure 4). Moreover, Dyslipidemia is characterized by increased Triglyceride, LDL-C, sd-LDL, and ox-LDL levels, with a decrease in HDL-C level. These findings are confirmed by a significant increase in the TG/HDL ratio in DEX compared to the control group (Figure 5 and Figure 6).

As depicted in Figure 4, in response to the injection of long-acting corticosteroids dexamethasone, there was a significant (*p* < 0.01) increase in both fasting glucose and insulin. Further, HOMA-IR increased significantly versus control. Feeding rats with LCHF significantly lowered the fasting glucose and insulin levels and reduced the HOMA-IR compared to DEX rats and even to control rats. On the other hand, HCLF significantly increased the fasting glucose and insulin compared to both control and DEX groups, increasing insulin resistance.

#### 3.3.2. LCHF Diet Decreased the Circulating Lipids

It is well known that dexamethasone-induced Dyslipidemia is associated with decreased HDL-cholesterol, increasing the plasma triglycerides level, and increasing the TG/HDL-C ratio, associated with atherosclerosis risk. As shown in the DEX group, Dyslipidemia was evident in our study, as noted from the significant increase in TG, with a significant decrease in HDL-C compared to control rats. While feeding, the HCLF diet produced harmful effects in rats, as shown by the significant (*p* < 0.05) increase in TG compared to control and DEX rats. Looking for the potential improvement in Dyslipidemia markers in response to LCHF, there was a significant decrease in TG and TG/HDL-C ratio, with a substantial increase in HDL-C versus DEX group, as depicted in Figure 5.

#### 3.3.3. The Impact of DEX, LCHF, and HCLF on the Levels of LDL-Cholesterol, OX-LDL, and sd-LDL

As depicted from Figure 6, compared to control rats, the DEX group showed a significant increase in LDL-cholesterol, OX-LDL, and sd-LDL. Feeding rats with LCHF decreased the circulating LDL-cholesterol significantly (*p* < 0.05), with insignificant changes in both OX-LDL and sd-LDL compared to the DEX group. Feeding rats with the HCLF diet significantly increased the oX-LDL and LDL-cholesterol (*p* < 0.05) compared to LCHF-fed rats.

#### 3.3.4. Changes in the Apo A, Apo and their Ratio in the Different Groups

In response to DEX, the rats showed a significant increase in the ratio between Apo B to Apo A lipoprotein due to the significant (*p* < 0.05) increase in Apo B and the drop of the Apo A lipoproteins. Restricting the carbohydrate intake and replacing with high-fat feeding to rats decreased the ratio significantly (*p* < 0.05) compared to DEX rats, and HCLF increased the ratio significantly (*p* < 0.05), compared to all groups (Figure 7).

#### 3.3.5. Correlation between sd-LDL, ox-LDL and HOMA-IR, TG/HDL Ratio, Apo B/Apo A Ratio

Correlation analysis revealed significant positive correlations between sd-LDL and HOMA-IR, TG/HDL ratio, and Apo B/Apo A ratio (r = 0.7955, *p* ≤ 0.0001, r = 0.7942, *p* ≤ 0.0001 and r = 0.911, *p* ≤ 0.0001, respectively) (Figure 8). Significant positive correlations were found between ox-LDL and HOMA-IR, TG/HDL ratio, and Apo B/Apo A ratio (r = 0.7246, *p* ≤ 0.0001, r = 0.7395, *p* ≤ 0.0001 and r = 0.667, *p* ≤ 0.0001, respectively) (Figure 9).

#### 3.3.6. Correlation between Apo B/Apo A Ratio and TG/HDL Ratio

The relationship between Apo B/Apo A ratio and TG/HDL ratio showed a significant positive correlation in all studied animal groups (r = 0.891, *p* ≤ 0.0001) (Figure 10).

## 4. Discussion

Metabolic Syndrome (MetS) is becoming more widespread, as a global health care problem and a socioeconomic burden [5,21]. It leads to significant health hazards in developed and developing nations [22]. The global prevalence of metabolic syndrome ranges from 16.7% by NCEP-ATP III Def. to 18.9 by IDF in the healthy population [23], and in adults, varies from 16% to 57% in Saudi Arabia, depending to the location [24], with a general MetS prevalence of 39.3% among Saudis aged between 30 and 70 years [25,26]. It is reported that the majority of metabolic syndrome and its consequent socioeconomic impact is linked to lifestyle behaviors, especially dietary habits [27]. Decreased insulin sensitivity and insulin resistance are associated with a high risk of cardiovascular disease, disturbed lipid, lipoprotein metabolism (Dyslipidemia), the subsequent overproduction of potentially atherogenic lipid and lipoproteins [28] and development of insulin resistance.

Therefore, the current study aimed to investigate the possible protective role of dietary intervention by using a low-carbohydrate/high-fat diet on metabolic-syndrome-associated Dyslipidemia in an experimental animal model.

The current study used dexamethasone to induce the metabolic syndrome model [13], and Dyslipidemia was a prominent component in the models. A significant increase was found in anthropometric and metabolic parameters, such as BMI, weight, glucose, insulin, HIMAIR, LDL, TG, sd-LDL, ox-LDLand Apo B, with a significant decrease in other parameters, including HDL and Apo A, compared to the negative control group. This result is consistent with the definition of most organizations for metabolic syndrome [12,28,29,30].

Dexamethasone (DEX) is a synthetic glucocorticoid, used in this study to induce the model of metabolic syndrome. The drug induces glucose intolerance, hyperglycemia, hyperinsulinemia, decreased muscle mass, and hepatomegaly [31]. Consequently, disturbed protein and lipid metabolism and serum lipid profile abnormalities are common effects [32]. Therefore, it is proved that it can be used for the induction of metabolic syndrome in an animal model.

Insulin resistance is evident in the current study by the significant increase in fasting blood glucose, insulin, HOMA-IR, atherogenic parameters (sd-LDL, ox-LDL, and Apo B/Apo A ratio). The role of Glucocorticoids can explain this to induce the insulin resistance state. In the insulin-resistant state, the alteration in LDL particles results in a predominance of small, dense LDL caused mainly by Hypertriglyceridemia. It leads to the atherogenic process. Our results confirm the results previously published by Ruotolo and Howard [33], Kappe et al. [34], Pandey et al. [35], and Kelsall et al. [36].

Moreover, in an insulin-resistance state, the LDL particles undergo many modifications that alter their size, density, and chemical properties within the blood flow and vascular wall [37]. The insulin resistance associated with free radicals oxidize LDL and produce oxidized LDL (ox-LDL) [38]. Oxidized LDL triggers the gene expression of adhesion molecules on the cell surface and, thus, endothelial dysfunction, resulting in progression to atherosclerosis [37]. In addition, Sierra-Johnson found that an increased apo B/apo AI ratio is significantly associated with insulin resistance in the non-diabetic US population [39]. In theory, the apoB100/apo AI ratio is a better predictor for assessing CHD risk than conventional markers [7]. This finding can be explained by the concentrations of plasma apoB100, reflecting the total quantity of atherogenic particles in VLDL, IDL-C, LDL-C, and lipoprotein (a) (Lp-a) and, the plasma content of apo-AI represents the total of antiatherogenic particles in HDL [7]. Thus, the apoB100/apo AI ratio might be a better marker for CHD progression, confirming its ability to expose the balance between atherogenic particles and antiatherogenic particles [40].

In the current study, an increase in Triglyceride/High-Density Lipoprotein-Cholesterol ratio (TG/HDL-C) is confirmative of dyslipidemia and insulin resistance in metabolic syndrome. It is reported that an abnormally high TG/HDL-c ratio was detected in 48% of metabolic syndrome subjects [41]. Consequently, it is considered a substitute marker for MetS [42] and a good indicator of insulin resistance [43,44].

Hepatic steatosis and nonalcoholic steatohepatitis (NASH) are considered the liver manifestation of metabolic syndrome [45]. At the histological level, the metabolic syndrome group showed evident accumulation of lipid droplets with disruption of the typical hepatic lobule architecture, indicating hepatic steatosis in our study. The hepatic steatosis in glucocorticoid-induced metabolic syndrome could be explained by the metabolic effect of chronic glucocorticoid administration in the form of hyperglycemia, hypertension, and hepatic steatosis. These effects ultimately result in insulin resistance and disturbed lipid metabolism due to the induction of the genes responsible for gluconeogenesis and hepatic lipogenesis [46].

Back to the primary purpose of the current study, feeding a low-carbohydrate high-fat Diet to rats administered dexamethasone with the same dose that induces metabolic syndrome in Group K revealed noticeable improvements at the level of anthropometric, histological, and biochemical parameters.

Compared to the metabolic syndrome model group, the low-carbohydrates-high-fat-diet-fed group showed significantly lower body weight, BMI, glucose, insulin, and HIMA-IR. These findings indicate that high-fat low-carbohydrate diets are associated with positive effects, such as weight loss, improved insulin sensitivity, and lower fasting blood glucose levels. These results are confirmative of the results in [2,4].

The enhancement of insulin sensitivity could explain the positive impact of low-carbohydrate high-fat diets on glycemic control, as it reduces visceral obesity, hence, the decrease in total body weight and BMI, increased insulin sensitivity, reduction in atherogenic Dyslipidemia and inflammatory biomarkers production [47].

Regarding the effects of a low-carbohydrate high-fat diet on metabolic-syndrome-associated Dyslipidemia, TG, LDL-C, sd-LDL, ox-LDL, Apo B/Apo A ratio, and TG/HDL ratio were significantly lower in Group K than those in the metabolic syndrome model group (Group D) and those fed with a high-carbohydrate low-fat diet (Group HC).

Tay et al. compared very-low-carbohydrate high-unsaturated/low-saturated-fat diets (LCD) with those of a high-unrefined-carbohydrate low-fat (HC) diet in type 2 diabetes (T2DM). They found that LCD had a marked drop in triglycerides plasma level and significantly improved lipid profile [48]. Pilis et al. [49] found that the LCD in healthy middle-aged men caused minor unfavorable outcomes in their lipid profile. The LCD markedly exaggerated the ketogenesis process and improved resting total blood cholesterol (TC) and HDL-C [49]. A meta-analysis of randomized controlled trials studies in 2018 compared the effect of low-fat to high-fat diets in overweight individuals without metabolic disturbances. They reported that subjects on a low-fat diet had lesser total cholesterol and LDL-C levels but significantly higher-level TG and a fall in HDL-C than those on a high-fat diet [50]. Another meta-analysis study stated that LCD resulted in an obvious increased LDL-C, which is not inconsistent with our results. Still, on the other hand, there was an increase in HDL-C and a more significant reduction in TG plasma level [51,52,53].

The positive effects of a low-carbohydrate high-fat diet on lipid profile and improvement of metabolic-syndrome-associated Dyslipidemia could be explained as follows: restricting carbohydrates with plenty of fat in the diet modifies the body’s endocrine response, which will induce a metabolic shift to use dietary fat, and the adipose-tissue-derived free fatty acids, for energy production rather than glucose. This effect is a normal body reaction to decreased plasma glucose and, consequently, diminished insulin to basal level, with increased adrenaline and glucagon blood levels. Then, the hepatic fatty acid supply for enhancement of the gluconeogenesis pathway increased. This process is associated with an increased rate of oxidated fatty acid and increased production of ketone bodies (acetoacetate, beta-hydroxybutyrate, and acetone). These water-soluble ketone bodies circulate freely in the blood and substitute glucose in tissues that usually require glucose, predominantly the brain. This effect is called nutritional ketosis. The body shifts from an insulin-mediated metabolism to another metabolic paradigm, with an increased ability to use fat as fuel [54], resulting in marked improvement in the lipid profile. Small dense LDL and apo B apoprotein are lowered with the LCHF diet and increased HDL-C and apo A1 [55]. These results agree with our study’s decreased apo B/apo A1 ratio in the HFLC-fed metabolic syndrome group.

KBs metabolism inhibits oxidative stress. Moreover, a reduced level of ox-LDL by LCHF is explained by the antioxidant effect of the associated ketone bodies produced by the nutritional ketosis, which protects the oxidation of LDL, with a reduction in ox-LDL level [56]. Β-Hydroxybutyrate (β-HOB) has been shown to reduce ROS production and improve mitochondrial respiration [57]. Moreover, the LCHF ketogenic diet stimulates the cellular endogenous antioxidant mechanisms by activating nuclear factor erythroid-derived 2 (NF-E2)-related factor 2 (Nrf2), the dominant antioxidant genes. This is in addition to upregulation of the transcription of antioxidant genes, such as catalase, mitochondrial superoxide dismutase, and metallothionein 2 [50,58].

The histological examination of the liver revealed marked improvement in the form of restoration of the hepatic lobule structure, with very little lipid droplets accumulation. This finding proves the antisteatotic effect of the LCHF (ketogenic) diet. The reduced amount of lipid droplets accumulation is an indicator of the reduction in intrahepatic triglyceride deposition by the impact of the nutritional ketosis, resulting from LCHF administration. The protective effect of the LCHF (ketogenic) diet against hepatic steatosis could be explained by the increased plasma level of non-esterified fatty acids, induced by the increased level of ketone bodies (KB). KD increases the plasma non-esterified fatty acid (NEFA) concentrations, the primary substrate of intrahepatic triglycerides (IHTG) [59]. These NEFA undergo either re-esterification into complex lipids as TG or transport into the mitochondria to be oxidized by β-oxidation into acetyl-CoA and link the TCA cycle for complete oxidation to CO_2_. Another important fate of acetyl CoA is the ketogenic pathway, as it is converted into acetoacetate (AAA) and β-hydroxybutyrate (β-OHB) [60]. Another important mechanism that explains the antisteatotic effect of the LCHF diet is the decreased glucose influx into the liver cells, with decreased insulin to the basal level. These inhibit hepatic gluconeogenesis glycogenesis lipogenesis [54,61,62].

The ketogenic diet induces hepatic VLDL receptor gene expression at the molecular level, leading to the inhibition of VLDL-mediated triglyceride hepatic release and its consequent decreased conversion to triglyceride-rich LDL. Moreover, the LCHF diet also increases lipoprotein lipase enzyme activity that mediates hepatic TG lipolysis [63]. In our study, these mechanisms explain why hepatic steatosis and TG/HDL ratio are reduced in LCHF-fed metabolic syndrome animals. The low-carbohydrate ketogenic diet induces hepatic VLDLR gene expression and promotes triglyceride clearance from VLDL in the liver, consequently protecting against liver steatosis progression. At the same time, the increased LPL activity is necessary for the clearance of triglycerides from VLDL in peripheral tissues [64]. The repressor effect of the LCHF diet on apo B gene expression was reported by Kirkpatrick et al. [52] in their review study. This finding could explain the decreased ApoB /apo A ratio detected in our study.

To elucidate the effect of the LCHF diet and confirm its protective effect, in all aspects of the studied parameters, a parallel group of rats fed with a conventional high-carbohydrate low-fat diet (HCLF) (Group HC) was also tested and investigated. By comparing the results of the HCLF-fed group of animals to those fed with HFLC, our hypothesis is more illustrated and confirmed. In response to the HCLF diet, which was isocaloric to the LCHF diet, there was a significant increase in fasting glucose, insulin, and HOMA-IR. Further, the serum TG increased and HDL-cholesterol decreased, with increased atherogenic markers. These data agreed with a previous study that showed a high-carbohydrate diet increased plasma triglyceride, small dense LDL, and reduced high-density lipoprotein (HDL) cholesterol with insulin resistance [65].

Regarding the extent of the clinical relevance of the current study, despite using an animal model of metabolic syndrome, it is proved that it showed the closest correlated criteria to metabolic syndrome in humans [66]. In our study, we chose the glucocorticoid-induced model of MetS. In clinical practice, it is reported that glucocorticoids are either endogenously produced in excess, as in the case of chronic stress or Cushing’s disease, or exogenous therapeutic glucocorticoids are associated with the development of metabolic syndrome [67].

Moreover, in clinical practice, the standard lipid profile for evaluating plasma lipids in an individual with metabolic syndrome under the LCHF diet is insufficient, as reported recently by Norwitz and Loh [68]. Therefore, the use of TG/HDL ratio and atherogenic parameters (sd-LDL, ox-LDL, and Apo B/Apo A ratio) for evaluating the efficacy of the dietary intervention on the use of the LCHF diet in the prevention and treatment of metabolic syndrome/Dyslipidemia is based on previous studies [7,38,43,44], as discussed above.

In addition, to the best of our knowledge, there are no published human clinical data about the effect of the LCHF diet on the apo B/apo A ratio in MetS, apart from a recent article in 2021 that was conducted on healthy young women with a normal body mass index [55].

## 5. Limitations

One of this study’s limitations is the inability to induce metabolic syndrome using high carbohydrates combined with a high-fat diet. Future studies will be conducted to explore the impact of LCHF on metabolic syndrome caused by diet. Further, this study’s measurement of ketone bodies is another limitation due to financial insufficiency.

## 6. Conclusions

The current study results concluded that Dyslipidemia is a crucial component of metabolic syndrome. Metabolic syndrome not only presents with increased Triglyceride-C and decreased HDL-C, but is also confirmed by a notable increase in apo B, small dense LDL, and oxidized LDL with a concomitant decrease in Apo A apoprotein. This metabolic-syndrome-associated Dyslipidemia is confirmed by the rise in both TG/HDL and apo B/ apo A ratio. The administration of a low-carbohydrate high-fat diet has a protective mechanism against this disorder. Its protective effect is proved at the biochemical level by improved TG/HDL and Apo B/apo A ratio that nearly returned to the normal levels. Moreover, reduction in the abnormal form of LDL-C (sd LDL and oxi-LDL) and the histological level in the reverse of metabolic syndrome associated with hepatic steatosis was also prominent. Further molecular research at the level of hepatic gene expression is recommended to explore the molecular mechanism of the ameliorative effect of a low-carbohydrate high-fat diet on metabolic-syndrome-associated Dyslipidemia.

## Figures and Tables

**Figure 1 nutrients-14-01903-f001:**
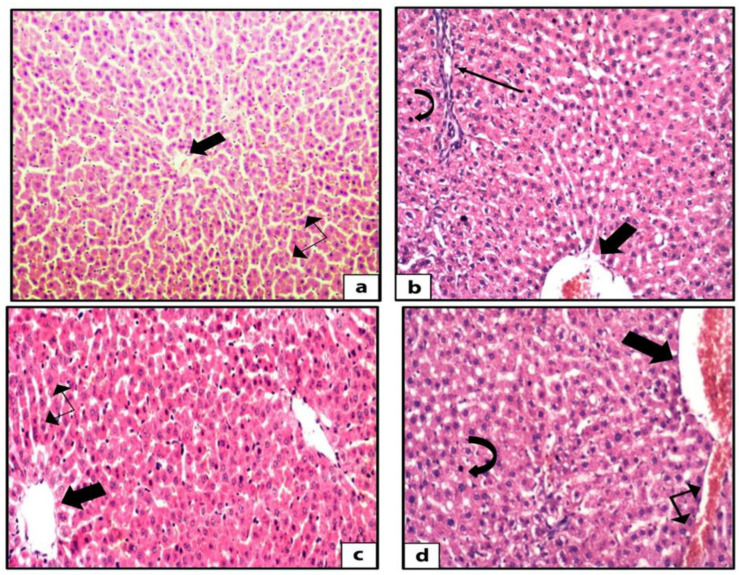
Photomicrographs of H&E-stained rat liver sections. The control rat liver shows branching and anastomosing cords of hepatocytes radiating from the central vein (thick arrow) with blood sinusoids (double head arrow) in between the cords (H&E ×200) (**a**). The sections of rat liver in DEX group showed disturbed arrangement of hepatocytes, multiple lipid droplets in between the hepatocytes, dilated congested central vein (thick arrow), mixed inflammatory infiltrate (thin arrow), and some hepatocytes with dark pyknotic nuclei (curved arrow) (H&E ×400) (**b**). The liver sections from DEX + LCHF group showed branching and anastomosing cords of normal hepatocytes radiating from the non-congested central vein (thick arrow), normal non-congested blood sinusoids (double-head arrow), and few lipid droplets (H&E ×400) (**c**). The sections of the liver of DEX + HCLF rats show multiple lipid droplets dilated congested central vein (thick arrow), dilated congested hepatic sinusoids (double-head arrow), and some hepatocytes with dark pyknotic nuclei (curved arrow) (H&E ×400) (**d**). DEX: rats treated with dexamethasone to induce metabolic syndrome; DEX + LCHF: rats treated with dexamethasone to induce metabolic syndrome and fed on low-carbohydrate high-fat diet; DEX + HCLF: rats treated with dexamethasone to induce metabolic syndrome and fed on high-carbohydrate low-fat diet.

**Figure 2 nutrients-14-01903-f002:**
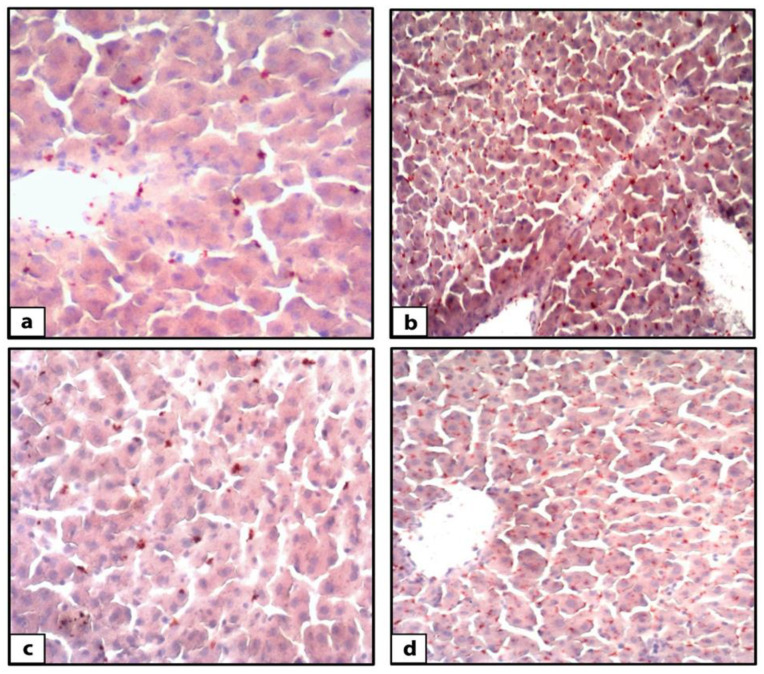
Photomicrographs of Sudan-III-stained rat liver sections. The control group (**a**) showed few lipid droplets (orange colored) accumulations in hepatocytes (Sudan III ×400). DEX-induced MetS group (**b**) showed an accumulation of lipid droplets (orange colored) in hepatocytes (Sudan III ×400). DEX+ LCHF group (**c**) showed few accumulations of lipid droplets (orange colored) in hepatocytes (Sudan III ×400). DEX + HCLF group (**d**) showed an accumulation of lipid droplets (orange colored) in hepatocytes. (Sudan III ×400). DEX: rats treated with dexamethasone to induce metabolic syndrome; DEX + LCHF: rats treated with dexamethasone to induce metabolic syndrome and fed on low-carbohydrate high-fat diet; DEX + HCLF: rats treated with dexamethasone to induce metabolic syndrome and fed on high-carbohydrate low-fat diet.

**Figure 3 nutrients-14-01903-f003:**
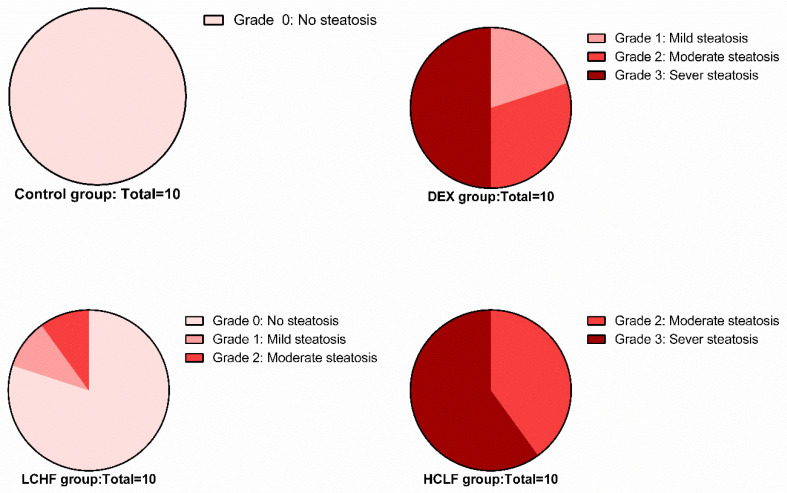
Pie chart shows the degree of steatosis in the control, DEX, DEX + LCHF, and DEX + HCLF groups. DEX: rats treated with dexamethasone to induce metabolic syndrome; DEX + LCHF: rats treated with dexamethasone to induce metabolic syndrome and fed on low-carbohydrate high-fat diet; DEX + HCLF: rats treated with dexamethasone to induce metabolic syndrome and fed on high-carbohydrate low-fat diet.

**Figure 4 nutrients-14-01903-f004:**
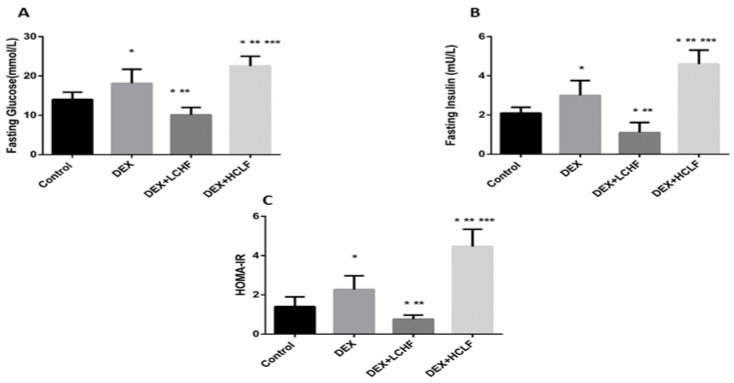
The impact of low-carbohydrate high-fat diet and high-carbohydrate low-fat diet on glucose homeostasis parameters in dexamethasone-treated rats. (**A**): fasting blood glucose; (**B**): fasting blood insulin, and (**C**): HOMA-IR. Data were expressed as mean + S.D. HOMA-IR: homeostasis model assessment of insulin resistance, DEX, DEX + LCHF and DEX + HCLF groups. DEX: rats treated with dexamethasone to induce metabolic syndrome; DEX + LCHF: rats treated with dexamethasone to induce metabolic syndrome and fed on low-carbohydrate high-fat diet; DEX + HCLF: rats treated with dexamethasone to induce metabolic syndrome and fed on high-carbohydrate low-fat diet. Statistically significant if *p* ≤ 0.05. *: Statistically significant compared to the Control group. **: Statistically significant compared to DEX group. ***: Statistically significant compared to DEX + LCHF group.

**Figure 5 nutrients-14-01903-f005:**
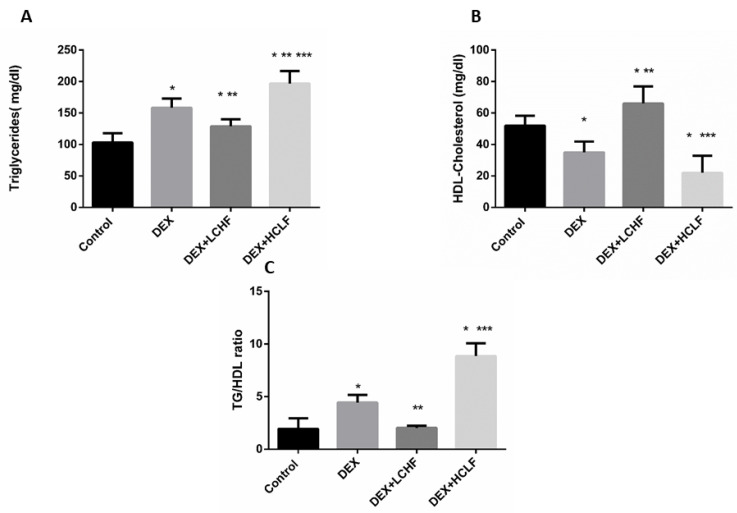
The impact of low-carbohydrate high-fat diet and high-carbohydrate low-fat diet on lipid profile parameters in dexamethasone-treated rats. (**A**): triglycerides; (**B**): HDL-Cholesterol, and (**C**): Triglycerides to HDL-Cholesterol ratio. HDL-Cholesterol: high-density lipoprotein-Cholesterol, DEX, DEX + LCHF, and DEX + HCLF groups. DEX: rats treated with dexamethasone to induce metabolic syndrome; DEX + LCHF: rats treated with dexamethasone to induce metabolic syndrome and fed on low-carbohydrate high-fat diet; DEX + HCLF: rats treated with dexamethasone to induce metabolic syndrome and fed on high-carbohydrate low-fat diet. Statistically significant if *p* ≤ 0.05. *: Statistically significant compared to the Control group. **: Statistically significant compared to DEX group. ***: Statistically significant compared to DEX + LCHF group.

**Figure 6 nutrients-14-01903-f006:**
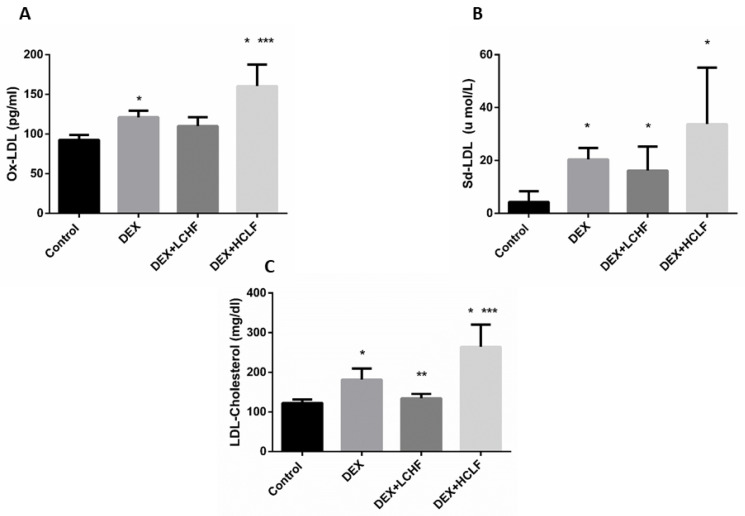
The impact of low-carbohydrate high-fat diet and high-carbohydrate low-fat diet on OX-LDL-Cholesterol (**A**), Sd-LDL-Cholesterol (**B**), and LDL-Cholesterol (**C**) in dexamethasone-treated rats. OX-LDL: oxidized low-density lipoprotein cholesterol; Sd-LDL: small dense low-density lipoprotein cholesterol; LDL-cholesterol: low-density lipoprotein cholesterol. DEX: rats treated with dexamethasone to induce metabolic syndrome; DEX + LCHF: rats treated with dexamethasone to induce metabolic syndrome and fed on low-carbohydrate high-fat diet; DEX + HCLF: rats treated with dexamethasone to induce metabolic syndrome and fed on high-carbohydrate low-fat diet. Statistically significant if *p* ≤ 0.05. *: Statistically significant compared to the Control group. **: Statistically significant compared to DEX group. ***: Statistically significant compared to DEX + LCHF group. LCHF diet modulates the glucose homeostasis in DEX-injected rats.

**Figure 7 nutrients-14-01903-f007:**
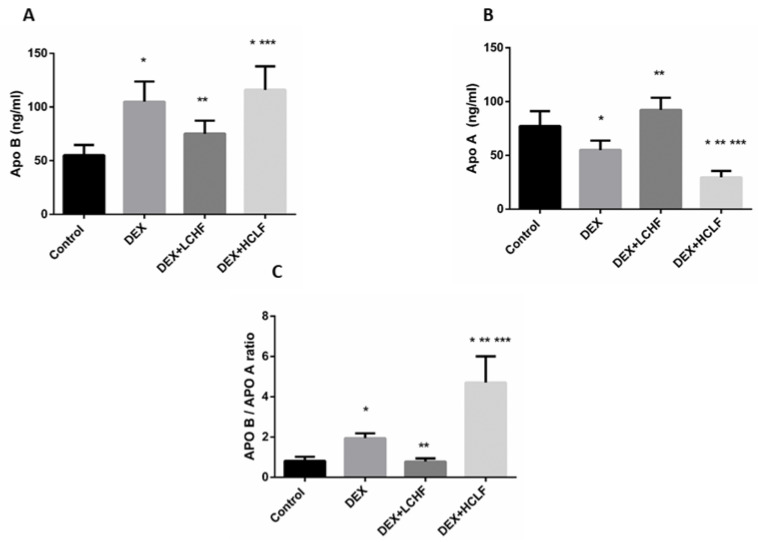
The impact of low-carbohydrate high-fat diet, and high-carbohydrate low-fat diet on APO B: (**A**) Apo A1 (**B**), and the APO B/ APO A1 ratio (**C**) in dexamethasone-treated rats. DEX: rats treated with dexamethasone to induce metabolic syndrome; DEX + LCHF: rats treated with dexamethasone to induce metabolic syndrome and fed on the low-carbohydrate high-fat diet; DEX + HCLF: rats treated with dexamethasone to induce metabolic syndrome and fed on high-carbohydrate low-fat diet. Statistically significant if *p* ≤ 0.05. *: Statistically significant compared to the Control group. **: Statistically significant compared to DEX group. ***: Statistically significant compared to DEX + LCHF group.

**Figure 8 nutrients-14-01903-f008:**
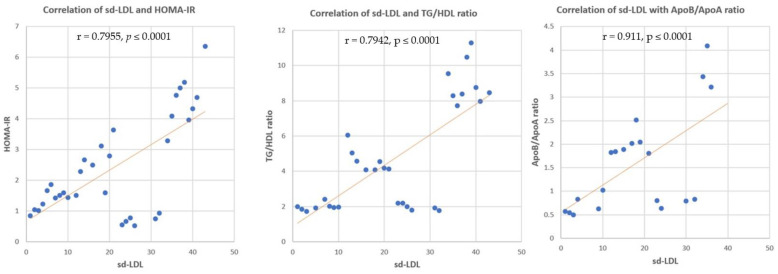
Correlation of sd-LDL and HOMA-IR, TG/HDL ratio, and Apo B/Apo A ratio. r = correla tion coefficient. *p* = value of statistical significance.

**Figure 9 nutrients-14-01903-f009:**
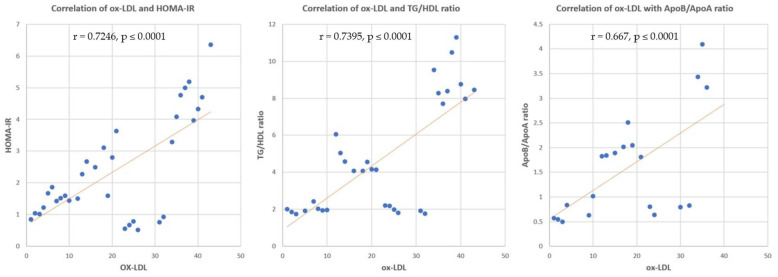
Correlation of ox-LDL and HOMA-IR, TG/HDL ratio, and Apo B/Apo A ratio. r = correlation coefficient. *p* = value of statistical significance.

**Figure 10 nutrients-14-01903-f010:**
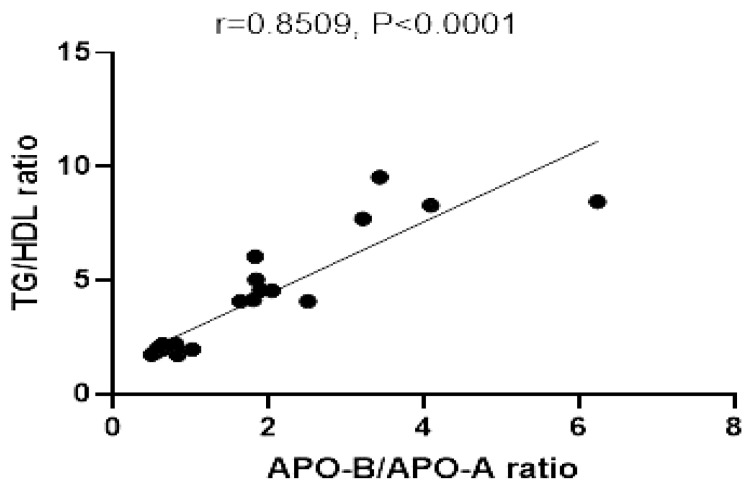
Correlation of TG/HDL ratio, and Apo B/Apo A. r = correlation coefficient. *p* = value of statistical significance.

**Table 1 nutrients-14-01903-t001:** The composition of Chow, LCHF, and HCLF diets.

	Chow	LCHF	HCLF
Carbohydrates	60%	5%	85%
Protein	28%	10%	5%
Fat	12%	85%	10%
Carbohydrates (gram/rat)	9	1	12
Protein (gram/rat)	5	2	1
Fat (gram/rat)	1	5.5	1
Energy (Calories/ rat)	61	61	60

**Table 2 nutrients-14-01903-t002:** The anthropometric parameters of all studied animal groups.

	Control	DEX	DEX + LCHF	DEX + HCLF
HIEGHT (B)	18.65 ± 0.51	18.45 ± 0.46	18.6 ± 0.75	18.58 ± 0.65
HIEGHT (D)	20.6 ± 0.35	20.7 ± 0.92	20.3 ± 0.72	20.95 ± 1.5 * ** ***
HIEGHT (POST D)	22.5 ± 0.59	22.7 ± 0.53	22 ± 0.27	23.8 ± 0.25
WIGHT (B)	133.4 ± 10.5	133.5 ± 10.2	133.5 ± 13.4	133.89 ± 10.6
WIGHT (D)	166.1 ± 10.8	179.67 ± 9.5 *	147.89 ± 3.50 **	201 ± 8.5 * ** ***
WIGHT (POST D)	200.4 ± 14.3	209.38 ± 11.90	126.17 ± 7.05 **	234 ± 14.19 * ** ***
BMI (B)	0.51 ± 0.031	0.52 ± 0.05	0.53 ± 0.05	0.51 ± 0.044
BMI (D)	0.65 ± 0.7	0.76 ± 0.08 *	0.51 ± 0.034 * **	0.93 ± 0.41 * ** ***
BMI (POST D)	0.72 ± 0.05	0.88 ± 0.05 *	0.35 ± 0.06 * **	0.97 ± 0.07 * ***

(B) measurement at baseline measurement. (D) measurement during induction with Dexamethasone. (Post D) measurement after stopping Dexamethasone. Data are presented as mean ± SD. SD: standard deviation, F for ANOVA test. Statistically significant if *p* ≤ 0.05. *: Statistically significant compared to Control group. **: Statistically significant compared to DEX group. ***: Statistically significant compared to DEX + LCHF group.

## Data Availability

The data presented in this study are available on request from the corresponding author.

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
