# Peer review of "Modulation of Dyslipidemia Markers Apo B/Apo A and Triglycerides/HDL-Cholesterol Ratios by Low-Carbohydrate High-Fat Diet in a Rat Model of Metabolic Syndrome"

_nutrients, 2022, doi:10.3390/nu14091903_

Round 1
Reviewer 1 Report
The current study results concluded that Dyslipidemia is a crucial component of metabolic Syndrome.
Metabolic Syndrome is not presented only by increased Triglyceride-C and decreased HDL-C only but also confirmed by a notable increase in apo B, small dense LDL, and oxidized LDL with a concomitant decrease in Apo A apoprotein.
This metabolic syndrome associated Dyslipidemia is confirmed by the rise in both TG/HDL and apo B/ apo A ratio.
The administration of a Low Carbohydrate High Fat Diet has a protective mechanism against this disorder.
Its protective effect is proved at the biochemical level by improved TG/HDL and Apo B/ apo A ratio that nearly returned to the normal levels.
Moreover, reduction of the abnormal form of LDL-C (sd LDL and oxi-LDL) and the histological level in the reverse of metabolic Syndrome associated with hepatic steatosis was also prominent.
Further molecular research at level of hepatic gene expression are recommended to explore the molecular mechanism of the ameliorative effect of Low-Carbo-hydrates High fat diet on metabolic syndrome-associated Dyslipemia
Author Response
Dear Editor-in-chief,
Dear Reviewers,
I am very glad to receive your letter, thanks for your time, effort, and support.
Please find below our point-by-point response for the manuscript nutrients-1712391. We have successfully replied to all the reviewer´s comments and revised the manuscript according to their precious comments. All changes were made in the revised version of our manuscript.
Yours;
Ayman Elsamanoudy
The corresponding author

Reviewer 2 Report
The paper was well done conducted e written.
Background, methods and materials and results are clearly described.
Authors should improve the explanation of the clinical relevance of data published, due to the study was conducted in an animal model.
Author Response
Dear Reviewers,
I am delighted to receive your letter. Thanks for your time, effort, and support.
Please find below our point-by-point response for the manuscript nutrients-1712391. We have successfully replied to all the reviewer´s comments and revised the manuscript according to their precious comments. All changes were made in the revised version of our manuscript.
Yours;
Ayman Elsamanoudy
The corresponding author
